# How Much Can CLIP Benefit Vision-and-Language Tasks?

**Sheng Shen**[*][†], **Liunian Harold Li**[*][‡], **Hao Tan**[°], **Mohit Bansal**[°],
**Anna Rohrbach**[†], **Kai-Wei Chang**[‡], **Zhewei Yao**[†] **and Kurt Keutzer**[†]
[†]University of California, Berkeley, [‡]University of California, Los Angeles
[°]University of North Carolina at Chapel Hill
{sheng.s, anna.rohrbach, zheweiy, keutzer}@berkeley.edu,
{liunian.harold.li, kwchang}@cs.ucla.edu, {haotan, mbansal}@cs.unc.edu

## Abstract

Most existing Vision-and-Language (V&L) models rely on pre-trained visual encoders, using a relatively small set of manually-annotated data (as compared to web-crawled data), to perceive the visual world. However, it has been observed that large-scale pre-training usually can result in better generalization performance, e.g., CLIP (Contrastive Language-Image Pre-training), trained on a massive amount of image-caption pairs, has shown a strong zero-shot capability on various vision tasks. To further study the advantage brought by CLIP, we propose to use CLIP as the visual encoder in various V&L models in two typical scenarios: 1) plugging CLIP into task-specific fine-tuning; 2) combining CLIP with V&L pre-training and transferring to downstream tasks. We show that CLIP significantly outperforms widely-used visual encoders trained with in-domain annotated data, such as BottomUp-TopDown. We achieve competitive or better results on diverse V&L tasks, while establishing new state-of-the-art results on Visual Question Answering, Visual Entailment, and V&L Navigation tasks.

## 1 Introduction

Vision-and-Language (V&L) tasks such as VQA (Antol et al., 2015) test a system's ability to understand and reason about the semantics of the visual world with the help of natural language. Most V&L models rely on a *visual encoder* to perceive the visual world, which translates the raw pixels into vectors from a representation space. Recent work (Anderson et al., 2018a; Jiang et al., 2020; Zhang et al., 2021) observes that the visual representation has become the performance bottleneck of V&L models and stress the importance of learning a powerful visual encoder. These high-performing visual encoders are trained on manually-annotated data with class labels (e.g., ImageNet) (Russakovsky et al., 2015) or bounding boxes (e.g., Visual Genome) (Krishna et al., 2017). However, such detection or image classification data is costly to collect, and the visual representation is limited by the pre-defined class labels. Thus, there is a need for a visual encoder that is trained on more diverse and large-scale data sources, unbounded by a fixed set of labels, and with generalization ability to unseen objects and concepts.

Recently, CLIP (Radford et al., 2021) has been proposed to learn visual concepts with language supervision. CLIP consists of a visual encoder and a text encoder. It is trained on 400M noisy image-text pairs crawled from the Internet. The data collection process is scalable and requires little human annotation. CLIP has shown strong *zero-shot* capabilities on benchmarks such as ImageNet classification. We hypothesize that it also bears great potential for the V&L tasks. However, directly applying CLIP as a zero-shot model to V&L tasks proves to be difficult (Section 5 and Kim et al. (2021)), as many V&L tasks require complex multi-modal reasoning. Thus, we propose to integrate CLIP with existing V&L models by replacing their *visual encoder* with CLIP's visual encoder.[1]

---

[*]The two authors contributed equally.

[1]Without confusion, we use the term CLIP to interchangeably refer to both the whole CLIP model (including the text and visual encoder) and just its visual encoder. We focus on studying CLIP as a visual encoder and provide analysis on CLIP's text encoder in Appendix A.6.

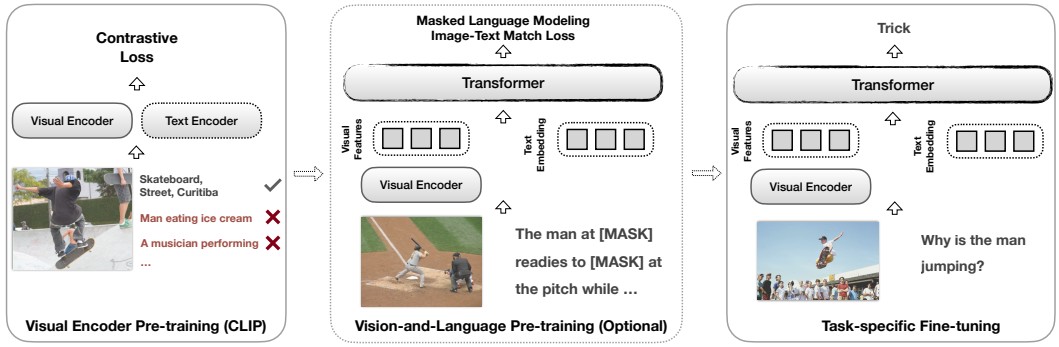

Figure 1: The training process of a V&L model typically consists of three steps: 1) visual encoder pre-training, 2) vision-and-language pre-training (optional), and 3) task-specific fine-tuning. In previous V&L models, visual encoder pre-training requires human annotated vision datasets, which is hard to scale up. Our CLIP-ViL proposes to use CLIP, which is trained on image-text pairs crawled from the Internet, as the visual encoder for V&L models. This reduces the need for human annotated in the pipeline and greatly improves model performance.

We present an empirical study on using CLIP as the visual encoder for diverse V&L tasks. We consider two typical scenarios: 1) we use CLIP in direct task-specific fine-tuning (Section 3); 2) we integrate CLIP with V&L pre-training on image-text pairs and transfer to downstream tasks (Section 4).[2] For clarity, we denote the models used in these two scenarios as **CLIP-ViL** (without V&L pre-training) and **CLIP-ViL$_p$** (with V&L pre-training).

In *direct task-specific fine-tuning*, we consider three widely-adopted tasks: Visual Question Answering (Antol et al., 2015), Image Captioning (Chen et al., 2015), and Vision-and-Language Navigation (Anderson et al., 2018b). On all three tasks, CLIP-ViL brings sizable improvement over strong baselines, 1.4% accuracy on VQA v2.0, 6.5 CIDEr on COCO Captioning, and 4.0% success rate on Room-to-Room navigation.

In *V&L pre-training*, we replace the conventionally used region-based representation (Anderson et al., 2018a) with CLIP. CLIP-ViL$_p$ performs exceptionally well on three benchmarks, including VQA v2.0, SNLI-VE (Xie et al., 2019), and GQA (Hudson and Manning, 2019), setting a new state-of-the-art (SotA) on VQA (76.70% on test-std), and SNLI-VE (80.20% on test). CLIP-ViL$_p$ with CLIP-Res50 outperforms models based on the widely used region-based encoder, BottomUp-TopDown (BUTD) ResNet101 (Anderson et al., 2018a). Moreover, CLIP-ViL$_p$ with CLIP-Res50x4 surpasses VinVL-ResNeXt152 (Zhang et al., 2021), which is an extreme scale-up attempt of the region-based encoder with human-annotated data.

## 2 BACKGROUND AND MOTIVATION

**Vision-and-Language (V&L) models.** V&L tasks require a model to understand the visual world and to ground natural language to the visual observations. Prominent tasks include visual question answering (Antol et al., 2015), image captioning (Chen et al., 2015), vision-language navigation (Anderson et al., 2018a), image-text retrieval (Wang et al., 2016) and so on. V&L models designed for these tasks often consist of a visual encoder, a text encoder, and a cross-modal interaction module (Kim et al., 2021).

We illustrate the three typical training stages in Figure 1: 1) the visual encoder is trained on annotated vision datasets (Russakovsky et al., 2015; Krishna et al., 2017) (denoted as *visual encoder pre-training*); 2) (optionally) pre-training on paired image-caption data with a reconstructive objective and an image-text matching objective (denoted as *vision-and-language pre-training*) (Lu et al., 2019); 3) fine-tuning on task-specific data (denoted as *task-specific fine-tuning*).

---

[2]We distinguish between *V&L pre-training* and *CLIP pre-training*: V&L pre-training models (Lu et al., 2019) have deep interactions between modalities while CLIP follows a shallow-interaction design (Section 2).

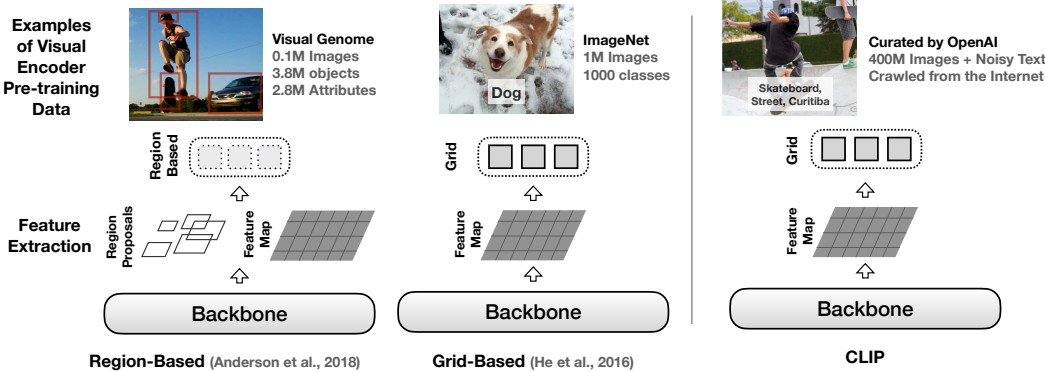

Figure 2: CLIP versus other visual encoders. Region-based methods (Anderson et al., 2018a) are trained on object detection data. For grid-based methods, previous work use either image classification (He et al., 2016) or detection data (Jiang et al., 2020). However, CLIP requires only aligned text.

**Visual encoders in V&L models.** Different models employ different visual encoders, we illustrate their architectures and pre-training processes in Figure 2. The encoders can be categorized as follows: 1) *region-based* models such as BUTD object detector (Anderson et al., 2018a; Kamath et al., 2021); 2) *grid-based* models such as Jiang et al. (2020) that directly extract grid-like feature maps from the visual backbone (He et al., 2016; Dosovitskiy et al., 2020).

The encoder is first pre-trained on human-annotated vision datasets. Region-based encoders are pre-trained with detection data such as Visual Genome (Krishna et al., 2017). Grid-based encoders are pre-trained with image classification data such as ImageNet (Russakovsky et al., 2015) or detection data (Jiang et al., 2020). However, these manually labeled datasets are expensive to construct and hard to scale up. They only provide supervision for a limited number of predetermined visual concepts. This motivates us to use CLIP as the visual encoder.

**CLIP.** CLIP (Contrastive Language-Image Pre-training) (Radford et al., 2021)[3] falls into the line of research that learns visual representations from natural language supervision (Desai and Johnson, 2020; Sariyildiz et al., 2020; Jia et al., 2021). CLIP follows a "shallow-interaction design", where a visual encoder and a text encoder encode an input image and text independently, and the dot-product between the two encoder's output is used as the similarity score between the input image and text. It is pre-trained with a contrastive loss where the model needs to distinguish aligned pairs from randomly sampled pairs. CLIP leverages an abundantly available source of supervision without human annotation: 400M image-text pairs found across the Internet. As a result, CLIP achieves SotA performance in a range of image classification and image-text retrieval tasks in a zero-shot setting.

## 2.1 MOTIVATION

Despite the strong zero-shot capability of CLIP on vision tasks, CLIP does not exhibit the same level of performance on certain V&L downstream tasks. For instance, if we cast VQA 2.0 (Goyal et al., 2017) into a zero-shot image-to-text retrieval task, we only observe chance performance (Section 5). Thus, we propose to integrate CLIP's visual encoder with previous V&L models (Figure 1). We consider the following CLIP variants with different visual backbones (He et al., 2016; Dosovitskiy et al., 2020) (CLIP-ResNet denoted as CLIP-Res): CLIP-Res50, CLIP-Res101, CLIP-Res50x4, CLIP-ViT-B/16 and CLIP-ViT-B/32. We next describe our methods in two scenarios: 1) direct task-specific fine-tuning (Section 3) and 2) V&L pre-training (Section 4).

---

[3]https://github.com/openai/CLIP

# 3   CLIP-VⅠL

In this section, we directly use CLIP as the visual encoder in task-specific models (referred as CLIP-ViL) and fine-tune on three representative tasks including Visual Question Answering (Section 3.1), Image Captioning (Section 3.2), and Vision-Language Navigation (Section 3.3).

## 3.1   VISUAL QUESTION ANSWERING

The task of Visual Question Answering (VQA) (Antol et al., 2015) is to provide the answer given an image and a related question. Various methods have been introduced (Fukui et al., 2016; Yang et al., 2016; Anderson et al., 2018a; Jiang et al., 2018; Gao et al., 2019; Jiang et al., 2020). Here, we select two representative approaches (i.e., Pythia (Jiang et al., 2018) and MCAN (Yu et al., 2019)) to study the impact of the CLIP visual encoders in VQA.

Table 1: Results on VQA v2.0. "†" marks results from (Jiang et al., 2020). CLIP visual encoders outperform all baselines, including strong visual encoders pre-trained with in-domain detection data (VG-* and BUTD-*).

| VQA Model | Visual Encoder | Result | |
|---|---|---|---|
| | | Test-dev | Test-std |
| Pythia | ImageNet-Res50† | 63.21 | - |
| | BiT$_M$-Res50 | 63.48 | 63.84 |
| | BiT$_M$-Res101 | 63.82 | 64.11 |
| | VG-ResNeXt-101† | 67.76 | - |
| | BUTD-ResNeXt-101† | **68.21** | - |
| | CLIP-ViT-B/32 | 59.14 | 59.56 |
| | CLIP-ViT-B/16 | 62.72 | 62.86 |
| | CLIP-Res50 | 65.55 | 65.78 |
| | CLIP-Res101 | 66.76 | 67.14 |
| | CLIP-Res50x4 | **68.37** | **68.68** |
| MCAN | ImageNet-ResNet50 | 67.23 | 67.46 |
| | BUTD-ResNeXt-101† | 72.01 | - |
| | VG-ResNeXt-101† | **72.59** | - |
| | CLIP-ViT-B/32 | 65.40 | 65.54 |
| | CLIP-Res50 | 71.49 | 71.72 |
| | CLIP-Res101 | 72.77 | 73.19 |
| | CLIP-Res50x4 | **74.01** | **74.17** |

**Experimental Setup.**   We evaluate on VQA v2.0 (Goyal et al., 2017) and follow Jiang et al. (2020)[4] for grid feature extraction. Details of Pythia and MCAN as well as full implementation details are included in the Appendix.

**Experimental Results.**   We report results on the VQA v2.0 Test-dev / Test-std set in Table 1. We compare with the following visual encoders:

- Standard ImageNet pre-trained visual encoders (**ImageNet-***);
- Visual encoders with SotA performance on ImageNet (**BiT$_M$-***) (Kolesnikov et al., 2020);
- Visual encoders pre-trained with detection data (**VG-*** and **BUTD-***) (Anderson et al., 2018a; Jiang et al., 2020).

Compared to the baselines, CLIP visual encoders demonstrate improvement. We especially note that VG-* and BUTD-* models are pre-trained on in-domain detection data, Visual Genome, which contain the sames images as VQA data. Thus, they significantly outperform baselines without such detection data (ImageNet-* and BiT$_M$-*). However, CLIP-* models without in-domain detection data can outperform VG-* and BUTD-*. Detection data are hard to scale up and contain limited object categories, while our results suggest training visual encoders on noisy image-text data as in CLIP is promising and scalable.

## 3.2   IMAGE CAPTIONING

Image captioning aims at generating a natural language description for an image. Various methods have been proposed for image captioning (Karpathy and Fei-Fei, 2015; Rennie et al., 2017; Anderson et al., 2018a; Luo et al., 2018; Luo, 2020). We investigate the effectiveness of the CLIP model for this popular task combined with the method proposed in Luo (2020).

**Experimental Setup.** We experiment with the basic Transformer model adapted from Vaswani et al. (2017) in Luo (2020). Grid feature maps are extracted for each image. We evaluate our model on COCO dataset (Chen et al., 2015). We use the standard automatic evaluation metrics including CIDEr (Anderson et al., 2016), BLEU (Papineni et al., 2002), METEOR (Lavie and Agarwal, 2007),

---

[4]https://github.com/facebookresearch/grid-feats-vqa

Table 2: Image Captioning results. B@4, M, C, and S are BLUE-4, METEOR, CIDEr and SPICE metric, respectively. "*" marks results from Luo (2020). CLIP-Res models outperform ImageNet pre-trained alternatives for both ResNet50 and ResNet101, as well as the strong in-domain region-based features from BUTD.

| Model | B@4 | M | C | S |
|---|---|---|---|---|
| BUTD (Anderson et al., 2018a) | 36.3 | 27.7 | 120.1 | 21.4 |
| VLP (Zhou et al., 2020) | 39.5 | 29.3 | 129.8 | 22.4 |
| AoANet (Huang et al., 2019b) | 38.9 | 29.2 | 129.8 | 22.4 |
| Oscar$_{base}$ (Li et al., 2020) | 40.5 | 29.7 | 137.6 | 22.8 |
| VinVL$_{base}$ (Zhang et al., 2021) | **40.9** | **30.9** | **140.4** | **25.1** |
| BUTD$_{Transformer}$* (Luo, 2020) | - | - | 127.7 | 22.5 |
| ImageNet-Res50$_{Transformer}$ | 36.2 | 27.6 | 118.8 | 21.2 |
| BiT$_M$-Res50$_{Transformer}$ | 37.4 | 28.1 | 122.7 | 22.1 |
| CLIP-Res50$_{Transformer}$ | 38.6 | 28.8 | 127.9 | 22.7 |
| CLIP-Res101$_{Transformer}$ | 39.2 | 29.1 | 130.3 | 23.0 |
| CLIP-Res50x4$_{Transformer}$ | **40.2** | **29.7** | **134.2** | **23.8** |
| CLIP-ViT-B/32 $_{Transformer}$ | 37.5 | 28.1 | 123.1 | 21.9 |
| CLIP-ViT-B/16$_{Transformer}$ | 39.8 | 29.5 | 133.2 | 23.4 |

Table 3: Unseen test results for Room-to-Room (R2R) dataset. 'SR' and 'SPL' are Success Rate and Success rate normalized by Path Length. 'Pre-Training' methods are mostly in-domain pre-trained on the Matterport3D (Chang et al., 2017) environments.

| Method | Unseen Test | |
|---|---|---|
| | SR | SPL |
| *No Pre-Training* | | |
| R2R (Anderson et al., 2018b) | 20 | 18 |
| RPA (Wang et al., 2018) | 25 | 23 |
| S-Follower (Fried et al., 2018) | 35 | 28 |
| RCM (Wang et al., 2019) | 43 | 38 |
| SMNA (Ma et al., 2019a) | 48 | 35 |
| Regretful (Ma et al., 2019b) | 48 | 40 |
| FAST-Short (Ke et al., 2019) | 54 | 41 |
| EnvDrop (Tan et al., 2019) | 51 | 47 |
| PRESS (Li et al., 2019b) | 49 | 45 |
| ALTR (Huang et al., 2019a) | 48 | 45 |
| CG (Anderson et al., 2019) | 33 | 30 |
| RelGraph (Hong et al., 2020) | 55 | 52 |
| **EnvDrop + CLIP-ViL** | **59** | **53** |
| *Pre-Training* | | |
| AuxRN (Zhu et al., 2020) | 55 | 50 |
| PREVALENT (Hao et al., 2020) | 54 | 51 |
| VLN-BERT(Hong et al., 2021)+OSCAR | 57 | 53 |
| VLN-BERT(Hong et al., 2021) | 63 | 57 |

and SPICE (Anderson et al., 2016). The scores are obtained on Karpathy test split (Karpathy and Fei-Fei, 2015) with beam search of 5 beams. Details are given in Appendix.

**Experimental Results.** We report Image Captioning results with different models in Table 2. Using the Transformer architecture from (Luo, 2020), we see that CLIP-Res models outperform ImageNet pre-trained alternatives for both ResNet50 (+9.1 / +1.5 in CIDEr / SPICE) and ResNet101 (+9.2 / +1.5 in CIDEr / SPICE). It even surpasses the strong in-domain region-based feature from BUTD and grid-based feature from BiT. As the model size grows in CLIP-ViL, the results also improve and the largest CLIP-Res50x4 achieves the best performance, although there still remains a gap to the pre-trained models that have interactive image-text pre-training phase like Oscar$_{base}$ and VinVL$_{base}$. Again, CLIP-ViT variant leads to worse performance compared to other visual modules, that we will discuss in Section 5.

## 3.3 VISION-AND-LANGUAGE NAVIGATION

Vision-and-language navigation tests the agent's ability to take action according to human instructions, which recently gains popularity in embodied AI (Anderson et al., 2018b; Chen et al., 2019; Jain et al., 2019; Chen et al., 2019; Qi et al., 2020b; Krantz et al., 2020; Nguyen and Daumé III, 2019; Ku et al., 2020). Specifically, the agent is put at a location in the environment (Chang et al., 2017) and asked to reach a target by following the language instructions. Here, we investigate the impact of the CLIP visual encoder on this new task.

**Model Architecture.** We experiment with the basic attentive neural agent as in Fried et al. (2018) (please refer to the original paper for implementation details). At each time step, the agent attends to the panoramic views and the instruction to make an action. We replace the pre-trained visual encoder from ImageNet pre-trained ResNet to the pre-trained CLIP visual encoders. Different from the VQA task that uses a feature map to include detailed information, we use a single-vector output for the entire image following previous works (Fried et al., 2018). For CLIP-ViT-B/32 models, we take the output of the [CLS] token. For CLIP-ResNet models, we take the attentive pooled feature (Radford et al., 2021) of the feature map. These features are also linearly projected and L2-normalized as in the CLIP model.

Table 4: Results of Room-to-Room (R2R) and Room-across-Room (RxR) datasets with original ResNet features and CLIP feature variants. 'BT-Agent' is the agent trained with back translation (BT). 'SR' is Success Rate. 'SPL' and 'nDTW' are the main metrics for R2R and RxR, respectively. The best results are bold. CLIP-ViL shows clear improvements over the previous ImageNet-trained ResNet model.

| Features | Room-to-Room | | | | Room-across-Room | | | | | | | |
|---|---|---|---|---|---|---|---|---|---|---|---|---|
| | Agent | | BT-Agent | | English | | Hindi | | Telugu | | Average | |
| | SR | *SPL* | SR | *SPL* | SR | *nDTW* | SR | *nDTW* | SR | *nDTW* | SR | *nDTW* |
| ImageNet-Res152 | 48.2 | 44.4 | 53.5 | 48.8 | 35.3 | 50.6 | 37.9 | 51.9 | 37.1 | 52.0 | 36.8 | 51.5 |
| CLIP-Res50 | 52.6 | 47.4 | 56.2 | 49.7 | 38.8 | 53.3 | 44.1 | 55.7 | 43.5 | 55.5 | 42.1 | 54.8 |
| CLIP-ViT-B/32 | 52.5 | 47.7 | 57.4 | 51.3 | 40.2 | 52.5 | 44.3 | 55.0 | 42.1 | 54.6 | 42.2 | 54.0 |
| CLIP-Res101 | 53.6 | 47.5 | 56.7 | 49.5 | **41.0** | 54.6 | **44.9** | **56.9** | 42.2 | 55.3 | **42.7** | 55.6 |
| CLIP-Res50x4 | **54.7** | **48.7** | **59.2** | **52.9** | 40.8 | **54.7** | 44.5 | 56.5 | **42.4** | **56.0** | 42.6 | **55.7** |

**Experimental Setup.** We apply our model to two vision-and-language navigation datasets: Room-to-Room (R2R, Anderson et al. (2018b)) and Room-across-Room (RxR, Ku et al. (2020)). R2R is built on the indoor environments from the MatterPort3D dataset (Chang et al., 2017). The environments are split into training, unseen validation, and unseen test. RxR extends the R2R dataset to multiple languages and follows the environment split. For R2R dataset, we follow the hyperparameter of the publicly available implementation[5] R2R-EnvDrop (Tan et al., 2019) and replace the input features[6] with the CLIP features. For RxR dataset, we change the path length and instruction length; details are given in Appendix.

Table 5: Unseen test results for Room-across-Room (RxR) dataset under mono-lingual setup. 'SR' and 'nDTW' are Success Rate and normalized Dynamic Time Warping.

| Method | Unseen Test | |
|---|---|---|
| | SR | nDTW |
| Random-Baseline (Ku et al., 2020) | 7.5 | 15.4 |
| Mono-Baseline (Ku et al., 2020) | 25.4 | 41.1 |
| SAA (Li et al., 2021a) | 35.4 | 46.8 |
| **EnvDrop + CLIP-ViL** | **38.3** | **51.1** |

**Experimental Results.** We show the test-unseen results of our best model (CLIP-Res50x4) and the comparison to the previous methods. On R2R dataset (in Table 3), CLIP-ViL reaches 8% higher in SR (success rate) and 6% higher in SPL (Success Rate normalized by Path Length) than our baseline, EnvDrop. CLIP-ViL outperforms previous non-pre-training agents and shows competitive results to VLN-specific pre-trained models. On RxR dataset (Table 5), CLIP-ViL achieves the best success rate and nDTW (normalized Dynamic Time Warping) under the mono-lingual setup (Ku et al., 2020) and is 4.3% better then the previous results for nDTW.

In Table 4, we compare different CLIP variants with the previous standard ResNet-152 feature extractors. These extractors are pre-trained on ImageNet and use the mean-pooled features as the representation for the image. CLIP-Res50 shows a clear improvement over the IN alternative ('ImageNet-Res152'). With larger models (i.e., 'CLIP-Res101' and 'CLIP-Res50x4'), the agent performance scales well on both R2R and RxR. Lastly, we find that the CLIP ViT model ('CLIP-ViT-B/32') has similar results as CLIP-Res50 model. ViT also shows a relatively better result when back translation (BT) is applied. The success of ViT model in VLN is possibly due to the use of [CLS] feature instead of the feature map.

## 4 VISION-AND-LANGUAGE PRE-TRAINING

Recently, V&L pre-training has been proposed as an effective technique to improve the performance on various V&L tasks (Lu et al., 2019; Tan and Bansal, 2019; Li et al., 2019a; Su et al., 2019; Chen et al., 2020; Zhou et al., 2020; Huang et al., 2020; Li et al., 2020; Zhang et al., 2021; Li et al., 2021b). Before task-specific fine-tuning, the model is pre-trained on aligned image-text data with a reconstructive objective and an image-text matching objective. We seek to test the potential of

---

[5]https://github.com/airsplay/R2R-EnvDrop
[6]https://github.com/peteanderson80/Matterport3DSimulator

Table 6: Evaluation results on three vision-and-language tasks. Our model with CLIP-Res50 outperforms most BUTD-based models. Our model with CLIP-Res50x4 sets a new state-of-the-art on VQA and SNLI-VE. It surpasses VinVL, which is a scaled-up version of BUTD and undergoes more intensive V&L pre-training than ours.

| Model | VisualEncoder | V&L Pretrain | | VQA | | SNLI-VE | | GQA | |
| | | Data | Epoch | Test-Dev | Test-Std | Dev | Test-P | Test-Dev | Test-Std |
|---|---|---|---|---|---|---|---|---|---|
| PixelBERT | ImageNet-Res50 | 5.5M | 40 | 71.35 | 71.42 | - | - | - | - |
| PixelBERT | ImageNet-ResX152 | 5.5M | 40 | 74.45 | 74.55 | - | - | - | - |
| LXMERT | BUTD-Res101 | 9.2M | 20 | 72.42 | 72.54 | - | - | 60.00 | 60.30 |
| UNITER | BUTD-Res101 | 6.5M | - | 72.70 | 72.91 | 78.59 | 78.28 | - | - |
| Oscar | BUTD-Res101 | 6.5M | 118 | 73.16 | 73.44 | - | - | 61.19 | 61.23 |
| VinVL | VinVL-ResX152 | 8.9M | 116 | 75.95 | 76.12 | - | - | **65.05** | **65.65** |
| **CLiP-ViL$_p$** | CLIP-Res50 | 9.2M | 20 | 73.92 | 74.09 | 78.64 | 78.97 | 59.79 | 60.55 |
| | CLIP-Res50x4 | 9.2M | 20 | **76.48** | **76.70** | **80.61** | **80.20** | 61.42 | 62.93 |

combining CLIP pre-training and V&L pre-training. We introduce CLiP-ViL$_p$, a vision-and-language model pre-trained on image-text data with CLIP visual encoder as its visual backbone. In the following, we introduce the model architecture and pre-training process of CLiP-ViL$_p$ in detail.

## 4.1 CLIP-VIL$_p$

**Model Architecture.** CLiP-ViL$_p$ assumes a text segment $T$ and an image $I$ as input. As in BERT, the text is tokenized into a sequence of subwords $\{w_1, w_2, ..., w_k\}$. Every subword is embedded as the sum of its token, position, and segment embeddings (Devlin et al., 2019) and thus the text is embedded as a sequence of word embeddings $\{\boldsymbol{w_1}, \boldsymbol{w_2}, ..., \boldsymbol{w_n}\}$. The image is embedded as a set of visual vectors $\{\boldsymbol{v_1}, \boldsymbol{v_2}, ..., \boldsymbol{v_m}\}$ from the grid-like feature map. The text and visual input are then concatanated into a sequence, $\{\boldsymbol{w_1}, \boldsymbol{w_2}, ..., \boldsymbol{w_n}, \boldsymbol{v_1}, \boldsymbol{v_2}, ..., \boldsymbol{v_m}\}$, and processed by a single Transformer. In most region-based models, the visual backbone is frozen as fine-tuning the object detector along with the Transformer remains an open problem (Su et al., 2019). In CLiP-ViL$_p$, the CLIP backbone is trained during both V&L pre-training and task-specific fine-tuning (see discussion in Section 5).

**Pre-training on Image-Text Data.** To learn unified representations for both vision and language, we follow prior work and pre-train the model on image-text pairs. We consider three pre-training objectives from LXMERT (Tan and Bansal, 2019): 1) grounded masked language modeling, where we randomly mask out 15% of words in the input sentence and train the model to reconstruct the masked words; 2) text-image matching, where the model is provided with a mismatched sentence with a probability of 0.5, and is trained to classify whether the text corresponds to the image; 3) visual question answering, where we train the model to predict the correct answer given a question.

## 4.2 EXPERIMENTS

**Setup.** We experiment with two variants of CLIP as the visual encoder, CLIP-Res50 and CLIP-Res50x4. Following LXMERT, we use the same corpora aggregated from MS COCO Captions (Chen et al., 2015), Visual Genome Captions (Krishna et al., 2017), VQA (Antol et al., 2015), GQA (Hudson and Manning, 2019), and VG-QA (Zhu et al., 2016) for pre-training. We follow the same pre-processing procedure and exclude any test data from the pre-training dataset. This results in 9.18M image-text pairs.

For computational efficiency, we use a relatively small resolution for images. We resize the shorter edges of images to 384 and the longer edges to under 640 with preserved aspect ratios. During pre-training, as the number of image patches is large, we randomly sample 100 image patches for every image following PixelBERT (Huang et al., 2020). We pre-train the model for 20 epochs and unfreeze the CLIP backbone during pre-training and fine-tuning. For details see the Appendix.

**Tasks.** For evaluation, we fine-tune the pre-trained model on three V&L tasks: VQA v2.0 (Goyal et al., 2017), visual entailment SNLI-VE (Xie et al., 2019), and GQA (Hudson and Manning, 2019). We provide more details in the Appendix.

Table 7: Zero-shot performance of CLIP on VQA v2.0 `mini-eval`, "PE" denotes we follow similar prompt engineering as suggested in CLIP paper.

| Model | VQA Question Type | | |
|---|---|---|---|
| | yes/no | number | other |
| CLIP-Res50 | 0.037 | 0.057 | 0.0 |
| CLIP-ViT-B/32 $_{PE}$ | 0.019 | 0.0 | 0.0 |
| CLIP-Res50$_{PE}$ | 0.055 | 0.057 | 0.0 |
| CLIP-Res101$_{PE}$ | 0.260 | 0.0 | 0.0 |
| CLIP-Res50x4$_{PE}$ | 0.446 | 0.118 | 0.034 |

Table 8: The importance of V&L pre-training (evaluated on VQA test-dev). All three models benefit from V&L pre-traibing significantly.

| Feature | No Pre-train | Pre-train | Diff |
|---|---|---|---|
| CLIP-Res50 | 64.66 | 73.92 | +9.26 |
| CLIP-Res50x4 | 69.91 | 76.48 | +6.57 |
| BUTD-Res101 | 66.70 | 72.42 | +5.72 |

**Results.** We report the results in Table 6. We include previous best pre-trained V&L models and their V&L pre-training data and epochs. As our model is based on BERT$_{BASE}$, we compare only with models based on BERT$_{BASE}$. The models are grouped by their visual encoder type. We first note that our two models perform competitively on all metrics. Especially, CLIP-ViL with CLIP-Res50x4 establishes a new SotA on VQA and SNLI-VE.

When comparing with the BUTD visual encoder trained on *in-domain data* (including LXMERT (Tan and Bansal, 2019), UNITER (Chen et al., 2020), and Oscar (Li et al., 2020)), our two models (CLIP-ViL with CLIP-Res50 and CLIP-Res50x4) significantly outperform most BUTD-Res101 based models. We especially note that LXMERT is trained on the same pre-training dataset and for the same number of epochs as our model, yet our CLiP-ViL$_p$ with CLIP-Res50 outperforms LXMERT on VQA by 2.59.

VinVL (Li et al., 2020) is an extreme scale-up of the region-based paradigm, which is pre-trained on multiple object detection datasets, including MS COCO (Lin et al., 2014), OpenImages (Kuznetsova et al., 2020), Object365 (Shao et al., 2019), and Visual Genome (Krishna et al., 2017). Yet, our model with CLIP-Res50x4 outperforms VinVL on VQA, while requiring significantly less steps of V&L pre-training. On GQA, our model under-performs VinVL. The potential reason is that GQA is automatically constructed from object bounding box data, which may give region-based models trained on such object data a significant advantage.

Lastly, we compare to Pixel-BERT (Huang et al., 2020), which takes a similar design as our model, but with an ImageNet initialized ResNet. CLIP initialization clearly holds advantage over ImageNet initialization, as CLIP-Res50 significantly outperforms Pixel-BERT with ImageNet-Res50.

## 5 ANALYSIS

In this section, we provide detailed analyses on a few interesting phenomena we observe during our experiments, which may help guide future exploration. Quantitative and qualitative analysis are provided to support our findings.

**Zero-Shot Performance of CLIP in VQA.** In the original paper, CLIP is intended as a zero-shot model and shows strong performance on various vision and image retrieval tasks. We are thus curious if CLIP can also perform well as a zero-shot model on V&L tasks that may require complex reasoning. To conduct zero-shot image classification, CLIP (Radford et al., 2021) uses the names of all classes in the dataset as the set of candidate text and predict the most probable (image, text) pair. We thus experiment with a similar setting on VQA but modify the candidate text to be the concatenation of question and answer pair for each question. Moreover, Radford et al. (2021) find a result improvement from prompt engineering. We follow this design by constructing "question: [question text] answer: [answer text]" as the prompt template. The results on VQA v2.0 `mini-eval` are shown in Table 7. All CLIP variants perform at near-chance level in the zero-shot setting while prompt engineering helps only a little. CLIP models also perform worse when the question becomes harder ("other" vs. "yes/no"). All these results suggest the need of a deep interactive model and additional pre-training/fine-tuning.

**Benefit of V&L Pre-training.** In Table 8, we compare the performance of models with or without V&L pre-training. We find that V&L pre-training brings significant performance improvement for the three models we test.

Interestingly, we find that CLIP models benefit more from V&L pre-training. Our conjecture is that the additional benefit could come from unfreezing the visual backbone. Because of technical difficulty in fine-tuning the object detector, most V&L models rely on frozen region-based encoders (Lu et al., 2019). But for grid-features such as CLIP, we can easily fine-tune the visual backbone and could potentially aid CLIP to adapt to the pre-training task. We hope that our finding inspires future work to further explore unfreezing the visual backbone in V&L models when computational budget allows.

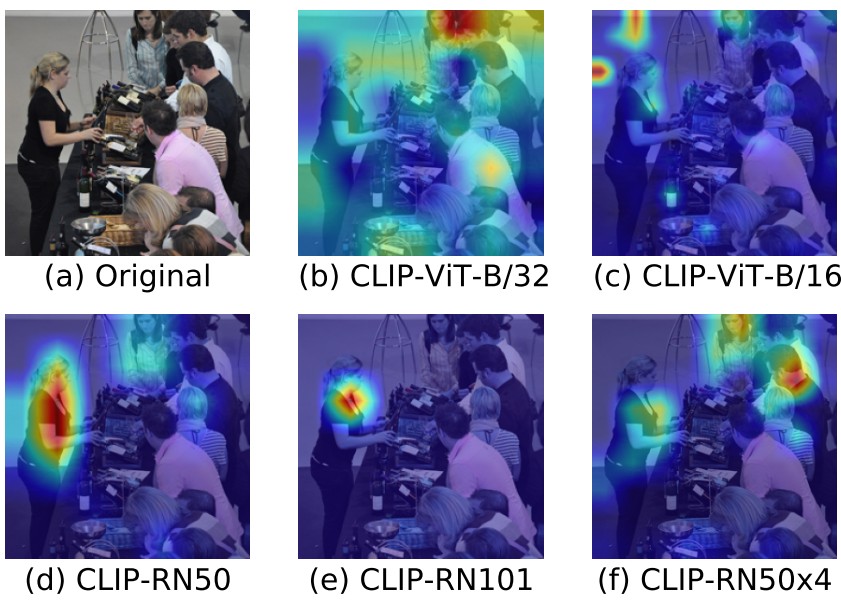

Figure 3: Grad-CAM Visualization of CLIP-ViT-B/32, CLIP-ViT-B/16, CLIP-Res50, CLIP-Res101 and CLIP-Res50x4 for the question "What color is the woman's shirt on the left?".

**Qualitative Comparison of CLIP Variants.** In our experiments, we find that ViT variants of CLIP under-perform their ResNet counterparts (Section 3.1). We perform Gradient-Based Localization (Grad-CAM) (Selvaraju et al., 2017) to visualize the salient regions idenfitied CLIP variants. We find that qualitatively, ResNet variants of CLIP localize objects better than ViT variants. For example, in Figure 3, CLIP-ResNet variants localizes the sentence "What color is the woman's shirt on the left?" better than CLIP-ViT variants without finetuning. This finding is inline with recent studies on vision transformers (Wang et al., 2021; Raghu et al., 2021; Dai et al., 2021). We provide more qualitative examples in the Appendix.

# 6 CONCLUSIONS

In this paper, we propose to leverage CLIP as the visual encoder for different V&L models across various tasks. We experiment with two approaches: in the first, we directly plug CLIP in task-specific fine-tuning; in the second, we integrate CLIP with V&L pre-training and fine-tune on downstream tasks afterwards. A variety of substantial experiments on different V&L tasks demonstrates that CLIP-ViL and CLIP-ViL$_p$ can achieve competitive or better performance as compared to strong baselines. Analyses from different perspectives explain certain intriguing phenomena and offer new directions for future V&L research.

## REPRODUCIBILITY STATEMENT

We provide the code to reproduce the main results in this paper in the supplementary material, which contains comprehensive instructions to reproduce our results. The code and model checkpoints will be made public.

## ACKNOWLEDGEMENT

We thank anonymous reviewers for their comments and suggestions. SS and KK were supported by grants from Samsung, Facebook, and the Berkeley Deep Drive Consortium. LL and KC were supported in part by DARPA MCS program under Cooperative Agreement N66001-19-2-4032. We would like to acknowledge DARPA, IARPA, NSF, and ONR for providing partial support of this work. The views expressed are those of the authors and do not reflect the official policy or position of the Department of Defense or the U.S. Government.

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

## A    APPENDIX

### A.1    VISUAL QUESTION ANSWERING

**Model Architecture**    Pythia encodes the question with an attention-based GRU (Chung et al., 2014) network and fuse the information with a multi-modal factorized bilinear pooling network. MCAN takes a LSTM (Hochreiter and Schmidhuber, 1997) as question encoder and an encoder-decoder based modular co-attention network for fusing multiple representations. Both models employ an output classifier on top of the fused representation to predict the final answer. To integrate CLIP for the VQA models, we extract grid features using CLIP. For CLIP-ViT-B/32 models, we reshape the patch representation from the final layer into grid features. For CLIP-ResNet models, we simply take the grid features from the last layer before the pooling.

**Implementation Details**    We follow (Jiang et al., 2020) to resize all input images to have a maximum shorter side of 600 pixels (longest 1000) when keeping the aspect ratio fixed. For training the detector on the VG dataset, we replace the backbone with CLIP visual module using implementation of Faster R-CNN in Detectron2[7]. For training VQA models, we use hyperparameters of the open-source implementation[8] from (Jiang et al., 2020) for the large version of the MCAN and base version of Pythia.

### A.2    IMAGE CAPTIONING

**Implementation Details**    For training, we follow the 'long epoch' hyperparameter of the publicly available implementation [9]. During the self-critical stage, we sample 5 captions for each image as in Luo (2020). For training objective, we experiment with the Self-Critical Sequence Training (SCST) in Rennie et al. (2017), where CIDEr (Vedantam et al., 2015) metric is optimized using REINFORCE algorithm (Williams, 1992).

### A.3    VISION-AND-LANGUAGE NAVIGATION

**Model**    For the model architecture, we experiment with the basic attentive neural agent as in Fried et al. (2018).

The agent model (i.e., another LSTM) then attends to the visual features and the language representations to predict the actions. At each time step $t$, the agent attends to the panoramic views $\{v_{t,i}\}_i$ and the instruction $\{w_j\}$ to make the action. The panoramic view is processed with a pre-trained

---

[7]https://github.com/facebookresearch/detectron2
[8]https://github.com/facebookresearch/mmf
[9]https://github.com/ruotianluo/self-critical.pytorch

Table 9: Comparison between grid features, CLIP features, and ImageNet-trained features on the R2R dataset. 'SR' and 'SPL' are success rate and success rate weighted by path length.

| Feature | Dimension | SR | **SPL** |
|---|---|---|---|
| ImageNet-Res152 | 2048 | 48.2 | 44.4 |
| CLIP-Res50 | 1024 | 52.6 | 47.4 |
| Grid-Res50 | 2048 | 47.6 | 44.7 |
| Grid-ResX101 | 2048 | 46.5 | 43.2 |
| Grid-ResX152 | 2048 | 47.8 | 44.6 |

visual encoder (e.g., ResNet) and the instructions are processed by a language LSTM (Hochreiter and Schmidhuber, 1997), denoted $\text{LSTM}_\text{L}$. The agent model, $\text{LSTM}_\text{A}$, then attends to the visual features and the language representations to predict the actions.

$$g_{t,i} = \text{ResNet}(v_{t,i}) \tag{1}$$

$$x_1, \ldots, x_l = \text{LSTM}_\text{L}(w_1, \ldots, w_l) \tag{2}$$

$$input_t = [\text{Attn}(h_{t-1}, \{g_{t,i}\}), \text{Attn}(h_{t-1}, \{x_j\})] \tag{3}$$

$$h_t, c_t = \text{LSTM}_\text{A}(input_t, h_{t-1}, c_{t-1}) \tag{4}$$

where $h_t$ and $c_t$ are the hiddens and states of the action LSTM at time step $t$, respectively. Please refer to Fried et al. (2018) for the implementation details.

**Implementation Details** We apply our model to two vision-and-language navigation datasets: Room-to-Room (R2R, Anderson et al. (2018b)) and Room-across-Room (RxR, Ku et al. (2020)). R2R is built on the indoor environments from the MatterPort3D dataset (Chang et al., 2017). The environments are split into training (61 environments), unseen validation (11 environments), and unseen test (18 environments). The agent is trained on the training environments (with 14,025 navigation instructions) and tested on separate sets of environments (2,349 in the unseen-validation and 4,173 in the unseen-test). RxR extends the R2R dataset with multiple languages and follow the environment split. Besides the multilingual nature, RxR is also more diverse in the navigation paths and richer in the present language. For R2R dataset, we follow the hyperparameter (e.g., batch size, learning rate, optimizer) of the publicly available implementation [10] R2R-EnvDrop (Tan et al., 2019) and replace the input features [11] with the CLIP features. To reduce the computational cost, the features are pre-extracted and frozen during the training of the navigational agent. For RxR dataset, we take the processed multilingual data provided in Li et al. (2021a) with Stanza tokenizers (Qi et al., 2020a). Since RxR dataset contains instructions longer than R2R, we change the maximum input length to 160 (from 80) and increase the imitation learning ratio from 0.2 to 0.4 to stabilize the training. Other training hyperparameters of RxR are the same as R2R. The models are trained on one RTX 2080 Ti GPU. It takes 1 days to converge in R2R and about 1.5 days to converge in RxR. We report two significant digits for R2R unseen test results following the leaderboard convention.

**Results Comparison to Grid Features** In the main paper, we compare the results regarding the ImageNet-pre-trained ResNet-152. We also report the comparison to grid features Jiang et al. (2020) that is trained with detection dataset. Jiang et al. (2020) showed that the results with these features are comparable to the original bottom-up attention with a heavy detection module. The same as the VQA task in Section 3.1, we test the performance of these detection-trained grid features on VLN tasks. Specifically, we use the mean pooling of the feature map as the representation of each view following previous works (Anderson et al., 2018b). As shown in Table 9, under the same ResNet50 backbone [12], we find that the detection-trained grid features are on par with the classification-trained grid features, still showing a gap to the contrastive-trained grid features. We hypothesize that the grid features inject regional knowledge into the dense feature map thus showing good results with grid-based modules (as shown in Section 3.1). However, pooling the feature map into a single feature vector (as in previous VLN works) leads to a loss of this dense information.

---

[10] https://github.com/airsplay/R2R-EnvDrop

[11] https://github.com/peteanderson80/Matterport3DSimulator

[12] The CLIP model uses an attention pooling module and makes modifications over the original ResNet (He et al., 2016) backbone.

## A.4    DETAILS OF CLIP-VIL$_P$

**Pre-training**    We pre-train with a batch size of 512. The Transformer is initialized from BERT$_{BASE}$ and optimized with an AdamW (Loshchilov and Hutter, 2017) optimizer. We use a linearly-decaying schedule and a peak learning rate of $1 \times 10^{-4}$ for the model with CLIP-Res50 and $5 \times 10^{-5}$ for the model with CLIP-Res50x4. The ResNet is initialized from CLIP and we use SGD with a learning rate of $3 \times 10^{-3}$. We decay the learning rate of SGD at epochs 12, 17 by a factor of 10. Per the suggestion of Tan and Bansal (2019), we only add the visual question answering loss during the later stage of the pre-training (the last 11 epochs) as the model is prone to overfit to the visual question answering loss. The model is trained on 8 Nvidia A100 GPUs and the pre-training takes around 5 days.

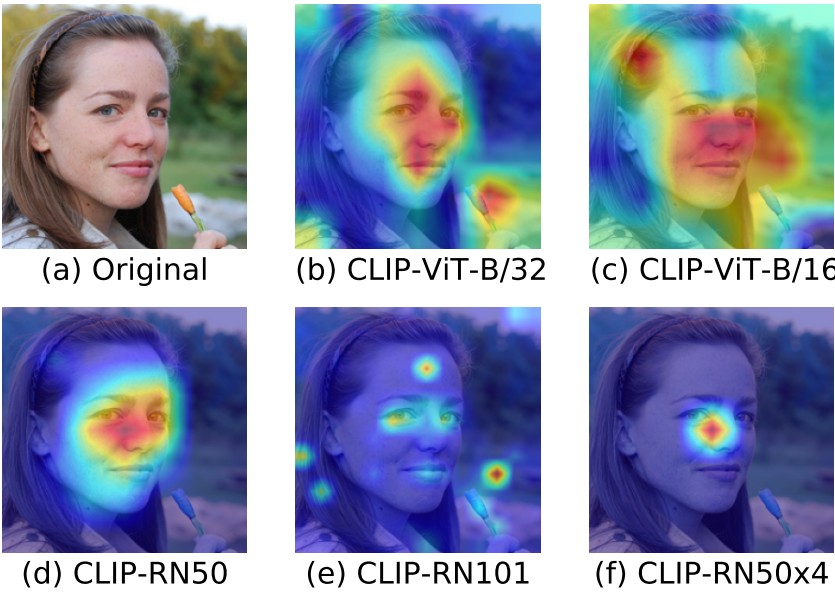

(a) Original    (b) CLIP-ViT-B/32    (c) CLIP-ViT-B/16

(d) CLIP-RN50    (e) CLIP-RN101    (f) CLIP-RN50x4

Figure 4: Grad-CAM Visualization of CLIP-ViT-B/32, CLIP-ViT-B/16, CLIP-Res50, CLIP-Res101 and CLIP-Res50x4 for the question "What color are her eyes?".

**Fine-tuning**    We fine-tune CLIP-ViL$_p$ on three tasks: VQA v2.0, SNLI-VE, and GQA. We introduce the task specifics and fine-tuning hyper-parameters in the following.

Every example in VQA consists of an image and a question, where the task is to predict the correct answer. We use the Karpathy split for training and validation (Karpathy and Fei-Fei, 2015). We fine-tune the model with the binary cross-entropy loss for 5 epoch with a batch size of 256. The Transformer is optimized with AdamW and a peak learning rate of $5 \times 10^{-5}$. The ResNet is optimized with SGD and an initial learning rate of $1 \times 10^{-3}$. We decay the learning rate of ResNet by a factor of 10 after epoch 3.

SNLI-VE is a three-way classification task, which involves determining the relation between an image and a sentence. The three possible relations include entailment, contradiction, and neutral. We fine-tune the model with the negative log-likelihood loss for 2 epoch with a batch size of 256. The Transformer is optimized with AdamW and a peak learning rate of $5 \times 10^{-5}$. The ResNet is optimized with SGD and an initial learning rate of $1 \times 10^{-3}$. We decay the learning rate of ResNet by a factor of 10 after epoch 1.

GQA follows the format of VQA but the questions and answers of GQA are automatically generated from ground-truth scene graphs. We use the same hyper-parameters as in VQA.

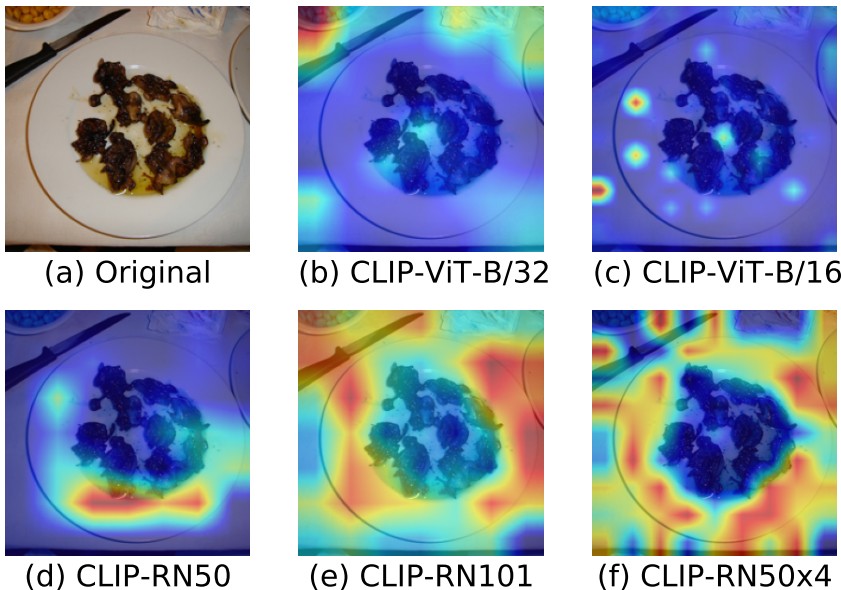

Figure 5: Grad-CAM Visualization of CLIP-ViT-B/32, CLIP-ViT-B/16, CLIP-Res50, CLIP-Res101 and CLIP-Res50x4 for the question "What is just above the plate?".

Table 10: The performance of finetuned CLIP text encoder and visual encoder without VLP on VQA.

| Image Encoder | Text Encoder | VQA$_{\text{mini-eval}}$ |
|---|---|---|
| CLIP-Res50 | BERT-base | 62.66 |
| | RoBERTa-base | 62.85 |
| | CLIP-Res50-text | 62.24 |
| CLIP-ViT-B/32 | BERT-base | 61.51 |
| | RoBERTa-base | 61.79 |
| | CLIP-ViT-B/32-text | 61.12 |

## A.5 MORE QUALITATIVE EXAMPLES

Here we present more qualitative examples using (Grad-CAM) (Selvaraju et al., 2017) to visualize the salient regions of CLIP models. Figure 4 and Figure 5 suggest that CLIP-ResNet localizes the sentence better than CLIP-ViT variants.

## A.6 ANALYSIS ON THE CLIP TEXT ENCODER

We extended the experiments in Table 8 (VQA$_{\text{test-dev}}$) without pre-training while fine-tuning both visual encoder and text encoder on VQA$_{\text{mini-eval}}$. The results suggest the CLIP text encoder consistently perform worse than BERT/RoBERTa counterparts even though CLIP is pre-trained with "in-domain" image-text pairs.

For directly analyzing the capability of the CLIP text encoder, we add experiments with fine-tuning CLIP text encoder on representative NLU tasks (GLUE). On the largest two tasks (QQP, MNLI), BERT-base (12 layer, 768 width) achieves $87.1 \pm 0.2$ on QQP and $77.9 \pm 0.3$ on MNLI. CLIP-Res50 text encoder (12 layer, 512 width) achieves $72.0 \pm 0.3$ and $51.6 \pm 0.4$. CLIP-ViT-B/32 text encoder achieves similar performance as CLIP-Res50 with the same architecture. CLIP-Res50x4 text encoder (12 layer, 640 width) achieves $73.8 \pm 0.3$ and $53.8 \pm 0.3$. We conduct the experiments with 32 batch size, 3 epochs, 3 random seeds and search the learning rate in $[1 \times 10^{-6}, 1 \times 10^{-5}, 3 \times 10^{-5}, 5 \times 10^{-5}]$. These results may directly reflect the inferior text encoder of CLIP which may be caused by the noisy and short text in the paired image-text data (Tan and Bansal, 2020).

