# OpenReview forum: "How Much Can CLIP Benefit Vision-and-Language Tasks?"
_ICLR.cc/2022/Conference — ICLR 2022 Poster_

### Official Review · Reviewer_EUMT · 2021-10-23

**Correctness:** 4
**Technical Novelty And Significance:** 3
**Empirical Novelty And Significance:** 3
**Recommendation:** 8
**Confidence:** 4

**Main Review:**

**Strengths:**

1. This work presents experiments on straightforward, but significantly important baselines for the vision-and-language community. The vision features have been shown to be a bottleneck of vision-and-language systems, and a significant part of recent progress can be attributed to better visual encoders. CLIP has been greatly impactful in the field, and the experiments presented in this paper are therefore a must-know to the community, and would be of interest to many.

2. The experiments presented in this work are solid, informative and representative.

3. The paper is clear and well written.

**Weaknesses:**

1. While the the experiments presented in this paper are of great importance to practitioners and researchers interested in vision-and-language, a shortcoming of this paper is the lack of novelty.

2. The authors experiment only with a subset of the publicly released CLIP models, and do not provide numbers for the largest available models. While OpenAI only very recently released more models, it would be great to see their performance in a future version of this paper.

**Other comments:**

CLIP ViT-B could refer to both the ViT-B/32 and ViT-B/16 architectures, and it would be great if authors clarified this in their manuscript.

**Summary Of The Paper:**

This paper explores how features from CLIP (Contrastive Language-Image Pre-training, Radford et al., 2021) affect the performance of vision-and-language models across a series of tasks. The authors explore using CLIP as a visual encoder in two settings, plugging its features directly into task-specific fine-tuning; and combining CLIP with intermediary vision-and-language pre-training before fine-tuning on downstream tasks. The experiments suggest that the simple change to CLIP offers significant benefits over commonly used encoders such as BottomUp-TopDown, providing very strong results across a wide range of vision-and-language tasks.

**Summary Of The Review:**

This work introduces a must-know baseline for vision-and-language research: using CLIP as a visual encoder. While this paper does not introduce a novel method, it provides important experimental information to many in the community. The results here presented would be of great interest to many, and I recommend it's acceptance to the conference.

---

> ### Author Response · Authors · 2021-11-22
> **Response to Reviewer EUMT**
>
> We thank the reviewer for reviewing our paper and are encouraged that the reviewer finds our experiments sufficient and significant.
>
> >1: more results for released OpenAI model.
>
> Thanks for suggesting more results regarding the recently released models from OpenAI. We just added the results for ViT-B/16 in VQA, captioning in Table 1,2 and Grad-CAM analyses in the updated manuscript, which shows the similar and interesting tendency of ViT features (please see our response for Reviewer ZXD1, Point 3.). We plan to include the results for all the released CLIP models in the final version as well.
>
> *We hope that our response and revision address your concerns and questions. We are happy to provide discussion if you have any further concerns or comments.*

---

### Official Review · Reviewer_ZXD1 · 2021-10-26

**Correctness:** 3
**Technical Novelty And Significance:** 3
**Empirical Novelty And Significance:** 3
**Recommendation:** 6
**Confidence:** 4

**Main Review:**

As a paper focusing on empirical studies, it has the following strengths:
+ The paper conducts extensive experiments to study the generalizability of CLIP pretraining across various tasks. The ablation studies on different types of visual encoders also provide insights into the application of pretraining models.

+ The proposed method is able to establish new state-of-the-art performance in multiple vision-and-language tasks, which can benefit future research in the related fields.

However, there are also some notable weaknesses:
- While I am convinced that CLIP could be beneficial for downstream tasks, it is still unclear to me what are the advantages of CLIP over existing V&L pretraining techniques and how does it achieve good results. For a purely empirical study on an existing technique,  I expect more in-depth analyses than simply showing a collection of results.

- After comparing CLIP with the other V&L pretraining techniques (e.g., Table 2 and Table 6), it seems that it is more advantageous only if used together with V&L pretraining. This is a bit tricky, because the V&L pretraining relies on 9.18M additional samples. It is unclear if the improvements achieved by the method truly result from the advantages of CLIP or are simply due to the use of more external data.

- Relating to the above comment, the analysis on unfreezing encoder is also problematic, as the comparison is between models without pre-training and with V&L pretraining. The higher accuracies of the latter ones could be attributed to the use of additional data, instead of the differences between frozen/fine-tuned visual encoders.

- It appears that CLIP only works well with grid-like features extracted from convolutional neural networks, i.e., no experiment on regional features and worse results when combined with a visual transformer. I am aware of the analyses on ViT-B, however, they only point out the defects of a single model. It does not answer the question of why CLIP can not be applied to different visual transformer models and features.


**Summary Of The Paper:**

This paper performs empirical analyses of applying the CLIP to various vision-and-language tasks. It demonstrates the potentials of the model in generalizing to different downstream applications, and provides some suggestions for model deployment. Experimental results show that CLIP pretraining leads to competitive performance and further combining it with V&L pretraining can outperform existing methods.

**Summary Of The Review:**

While I have no doubt that CLIP could offer opportunities for developing new state-of-the-art models, this paper falls short of explaining the reasons behind the successes of CLIP. I also found several experiments and analyses problematic and do not quite support the claims. Overall, this is a borderline paper to me as it does demonstrate the potential of CLIP with good results. Therefore, I am slightly leaning towards accepting it.

---

> ### Author Response · Authors · 2021-11-22
> **Response to Reviewer ZXD1**
>
> We thank the reviewer for their feedback and suggestions.
>
> >1. “what are the advantages of CLIP over existing V&L pretraining techniques”
>
> Our main claim is that we can power V&L models with CLIP as the visual backbone. Therefore, our experiment focuses on **comparing various visual encoders**. This is in an orthogonal direction with existing V&L pre-training techniques. V&L pre-training focuses on training a deep-fusion model with image-caption data for high-level V&L tasks involving complex reasoning. V&L pre-training methods reply on a visual feature encoder to extract low-level visual features. Prior work uses object detection based visual encoder trained on human annotated data. In our paper, we show that CLIP is a strong alternative visual encoder trained on noisy web data.
>
> We do not show that CLIP holds advantage over existing V&L pre-training techniques, but we show that CLIP holds advantage over previous visual features used in V&L pre-training. Thus, we show **CLIP can be combined with prior V&L pre-training** techniques (Table 6) to bring large improvements.
>
> >2: “ (Table 2 and Table 6) … unclear if the improvements achieved by the method truly result from the advantages of CLIP or are simply due to the use of more external data …  the analysis on unfreezing encoder is also problematic, as the comparison is between models without pre-training and with V&L pretraining”
>
> Table 1, 2, and 6 are separate tables. We are comparing CLIP against **existing visual features (e.g., BUTD)** under two scenarios:
> no V&L pre-training (thus with no extra data) (Table 1 & 2) and with V&L pre-training (thus with extra data) (Table 6).
>
> Within Table 6, all listed models use external V&L pre-training data. In particular, CLIP-ViL and LXMERT use exactly the same V&L pre-training data,  and we show that CLIP-Vil outperforms LXMERT. This comparison shows the effect of CLIP, when the baseline and our approach use the same V&L pre-training data.
>
> Within Table 1 or 2, all models have no extra V&L pre-training data. We can observe that CLIP visual features have an advantage over existing bases such as VG-* and BUTD-*.
>
> For analysis regarding unfreezing encoder (Table 8), please see our response for Reviewer Rnvh, Point 5.
>
>
> >3: why does the visual transformer model of CLIP exhibit worse performance?
>
> We thank reviewer Rnvh and ZXD1 for discussion towards the worse performance of CLIP visual transformer variants. We speculate that this may be attributed to the worse localization performance of the vanilla visual transformer as pointed out by reviewer Rnvh in [1, 2, 3].
>
> Qualitatively, we add more Grad-CAM results to visualize the localization performance of **CLIP-ViT-B/16**, CLIP-ViT-B/32, **CLIP-Res50, CLIP-Res101 and CLIP-Res50x4 (added analysis in Sec 5.)**. Beyond the two basic models discussed in the original paper, we hope these extended results intuitively show that CLIP-ResNet variants localize the concepts/objects in the sentence better than CLIP-ViT variants.
>
> Quantitatively, we added the results and briefly discussed the Low Detection Performance of CLIP-ViT variants (in Sec 5). On Visual Genome, we found that using the same architecture with replaced visual backbone, the Average Precision (AP) is only **0.03 for CLIP-ViT-B/32 and 0.17 for CLIP-ViT-B/16**, which is much lower than **1.05** for its **CLIP-Res50** alternative.
>
> We hope that our response and revision address your concerns and questions. We are happy to provide discussion if you have any further concerns or comments.
>
>
> [1] Do Vision Transformers See Like Convolutional Neural Networks? arxiv 2021 \
> [2] Dynamic Head: Unifying Object Detection Heads with Attentions, CVPR 2021 \
> [3] Pyramid Vision Transformer: A Versatile Backbone for Dense Prediction without Convolutions, ICCV 2021
>
> *We hope that our response and revision address your concerns and questions. We are happy to provide discussion if you have any further concerns or comments.*

---

> > ### Comment · Reviewer_ZXD1 · 2021-11-26
> > **Thanks for the response**
> >
> > I appreciate the authors for answering my questions and providing the additional results.
> >
> > I have no doubt that CLIP can serve as a strong visual backbone for V&L applications. My concerns are more about the underlying reasons behind the improvements of CLIP. The new results on BUTD with  V&L pretraining seem to indicate that the advantages of CLIP mostly come from the use of additional data instead of its contrastive learning paradigm, i.e., the relative gain from the model design is much smaller than that from additional data. I believe it is necessary to conduct similar experiments with other V&L pretraining methods. In this way, we may find the key factors behind the success of CLIP.
> >
> > I agree with the other reviewers that this paper introduces a must-known baseline to the V&L community. However, I still feel like the analyses in the paper are relatively shallow, e.g., lack of comparisons between different V&L pretraining methods and no analysis on the reasons behind CLIP’s improvements. While having a larger pool of powerful baselines is beneficial for downstream applications, it is more important to learn from these baselines and develop new methods based on the findings. Therefore, I decided to keep my original rating.

---

### Official Review · Reviewer_jbK4 · 2021-10-31

**Correctness:** 4
**Technical Novelty And Significance:** 1
**Empirical Novelty And Significance:** 3
**Recommendation:** 5
**Confidence:** 5

**Main Review:**

The tendency that more powerful visual encoders yield more performant VL models has been discussed and demonstrated in numerous papers (from the BUTD [^1] to VinVL [^2]), which emphasized the need for powerful visual backbones.
As CLIP showed the great generalization power of their visual encoders compared to the encoders trained with imagenet classification, I feel no surprise they boosted the performance of VL tasks compared to their imagenet pre-trained counterpart.
I must credit the paper for solidifying this tendency, but I am doubtful whether this paper opens any new directions to the community.
All experiments are replications of existing baselines, just switching the backbone to publicly shared CLIP weights.
I think not many insights besides "switching the visual encoder to CLIP's bring the performance boost" are given.
I was hoping the paper contains some analysis of CLIP's text encoder since the language side of VL models at least take something from what the CLIP's text encoder has learned, but sadly, I found only plug-and-plays of CLIP's visual encoders.
To raise my recommendation, please clarify what directions the paper could suggest to the community other than the general tendency I've mentioned.

[^1] Anderson, Peter, et al. "Bottom-up and top-down attention for image captioning and visual question answering." _Proceedings of the IEEE conference on computer vision and pattern recognition_. 2018.
[^2] Zhang, Pengchuan, et al. "Vinvl: Revisiting visual representations in vision-language models." _Proceedings of the IEEE/CVF Conference on Computer Vision and Pattern Recognition_. 2021.

**Summary Of The Paper:**

This paper examines how well CLIP's visual encoders are transferred on vision-and-language (VL) tasks.
The paper conducted three without-VLP tasks: VQA, image captioning, and vision-and-language navigation, and three with-VLP tasks: VQA, SNLI-VE, and GQA to show CLIP's visual encoders' transferability,
CLIP's resnet-based visual encoders consistently outperformed their Imagenet pre-trained counterpart.
The authors found that CLIP's ViT-based visual encoder performed far worse than the resnets.
From Grad-CAM and detection fine-tuning experiments, the authors speculated that ViT's features lack localization information.

**Summary Of The Review:**

Not many insights besides "switching the visual encoder to CLIP's bring the performance boost" are given.
To raise my recommendation, please clarify what directions the paper could suggest to the community other than the general tendency I've mentioned.

---

> ### Author Response · Authors · 2021-11-22
> **Response to Reviewer jbK4**
>
> We thank the reviewer for reviewing and appreciating that our results are solid. Below we address some of the concerns raised.
>
> >1. “whether this paper opens any new directions to the community”
>
> Thank you for raising this concern. We believe our papers bring in the following two key contributions that may not be known by the current literature:
>
> - **Our work revisits the direction of using grid features without detection data, which opens up a promising alternative to scaling up the visual encoder**
>
>      It may not be a surprise that CLIP outperforms its IN pre-trained counterpart, but it is a novel finding that it could outperform visual encoders trained with object detection data. This is a key difference. In Table 1 (we have revisited the presentation), we are not only comparing CLIP with iN pre-trained counterparts (IN-\*) but with strong visual encoders pre-trained with object detection data (VG-\* and BUTD-\*).
>
>      Object detection data have been reported to bring significant performance boost to V&L modeling (BUTD). Since the introduction of BUTD, detection-based visual features have been dominating the VQA leaderboard (all VQA winners since 2018 use detection data). Recent progress on visual features for V&L modeling is also attributed to the inclusion of detection data (Jiang et al. and VinVL). Among published works, we do not see strong evidence that a visual encoder without any detection data could outperform a visual encoder trained with detection data for V&L tasks.
>
>      Our work points out a different yet promising route than the established research line of BUTD: scaling up noisy semantic rich image-text data could deliver strong performance for V&L modeling. This could have a profound impact on the common practices in the V&L field. Detection data are hard to scale up and have limited object categories, while pre-training with noisy image-text data as in CLIP is promising and scalable.
>
>
> - **We report a “must-know” baseline to the community**
>
>      The community values the exploration and reporting of **important** baselines. CLIP has been greatly impactful in the field, and the experiments presented in this paper are therefore a must-know to the community and would be of interest to many.
>
>      In addition, it is not always straightforward to apply a visual encoder to a V&L task and how to scale up the visual encoder in V&L modeling is still under-explored. The statement “more powerful visual encoders yield more performant VL models” is not always accurate. For example, Table 20 in VinVL. shows that VinVL-4sets (a “more powerful” visual encoder) is trained with 20 times more image data than VinVL-VG but performs much worse. Such results call for the careful diagnosis of visual features in VL tasks, and it is important to establish must-know baselines with concrete experimental results.
>
>
> >2. “analysis of CLIP's text encoder”
>
> Thanks for suggesting more analyses regarding CLIP text encoder. We include more analyses (discussed below) in Appendix A.6 and plan to include more in our final version.
>
> - **Zero-shot evaluation of CLIP on VQA**
>
>      For analyzing CLIP text encoders without fine-tuning, we did the zero-shot evaluation of CLIP on VQA (in Sec. 5). The chance performance may suggest the direct plug-in of pre-trained CLIP text encoder is not good.
>
> - **CLIP’s text encoder perform worse on VQA than BERT/RoBERTa**
>
>      We finetune the models from Table 8 with different text encoders on VQA (w/o V&L pre-training and report on mini-eval). The results show the CLIP text encoder consistently performs worse than BERT/RoBERTa counterparts though CLIP is pre-trained with image-text pairs. This also motivates us to use BERT to initialize our CLIP-ViLp model.
>
>      >| Image Encoder | Text Encoder       | VQAv2 mini-eval |
> |---------------|--------------------|:-------------:|
> |   CLIP-Res50  | BERT-base          |     62.66     |
> |               | RoBERTa-base       |     62.85     |
> |               | CLIP-Res50-text    |     62.24     |
> | CLIP-ViT-B/32 | BERT-base          |     61.51     |
> |               | RoBERTa-base       |     61.79     |
> |               | CLIP-ViT-B/32-text |     61.12     |
>
>      Our conjecture is that pre-training data of CLIP may consist of a lot of short and noisy phrases. Thus the text encoder of CLIP may not do well on V&L tasks such as VQA, which require understanding of long and complex sentences. We provide below experiments to elaborate our hypothesis.
>
> - **CLIP’s text encoder performs poorly on NLP tasks**
>
>      We fine-tune CLIP’s text encoder on NLP tasks (GLUE), to evaluate it on longer and more complex sentences. We find the CLIP’s text encoder lags significantly behind BERT on two largest tasks: 13% drop on QQP and 24% drop on MNLI. This elaborates our hypothesis regarding CLIP’s text encoder for long and complex sentences. See details in Appendix A.6.
>
> *We are more than happy to provide discussion if you have any further questions!*

---

> > ### Comment · Reviewer_jbK4 · 2021-11-22
> > **Thank you for your response**
> >
> > I appreciate the authors' work on the CLIP's text encoder, and I enjoyed reading the authors' analysis in the updated appendix.
> >
> > I agree that the transferability from a detection-free pre-trained visual encoder to vision and language pre-training is under-explored. Though recent studies like ALBEF [^1], SimVLM [^2], or METER [^3] demonstrate the general tendency of how scaled-out visual encoders trained without object detection supervision can help the vision and language pre-training out, but I think it is a bit unfair to bring them in the discussion because they are concurrently done with this paper.
> >
> > I am certain that this paper reports a "must-know" baseline to the community, but I still believe the benefits to the community is limited; for example (sorry for bringing the concurrent work), ALBEF showed that the visual encoder (ViT-B/16) pre-trained with ImageNet-1K (DeiT regime) can still perform 75.84 VQA test-dev score after pre-trained with 14M image-text pairs. And I believe the community can easily adopt the modules and schemes ALBEF proposed (ITC and momentum distillation) since they do not require big, proprietary data. However, CLIP-ViL suggests the community rely on the CLIP pre-training weights, and it will not be an easy choice for someone who wants to use different visual encoders other than those OpenAI released.
> >
> > Though I still believe the impact of the main finding: "CLIP performs better on VL tasks compared to other visual encoders" is neither significant nor novel, I am raising my score to borderline reject as the paper reports "must-know" empirical results and other interesting things such as the analysis on the CLIP's text encoder or the worse performance of CLIP-ViT compared to the CLIP-ResNet (though I believe it is more of an issue of CLIP rather than of a ViT itself since ALBEF's ViT performs well on VL tasks.)
> >
> > (+) I'm not saying that all works dealing with proprietary data are insignificant. The papers like CLIP or ALIGN have their contributions which guide someone who wants to train the model on a large-scale dataset as training a model on a huge dataset is generally difficult. However, this paper has no such contributions as it just borrows open-sourced pre-trained weights.
> >
> > [^1] Li, Junnan, et al. "Align before fuse: Vision and language representation learning with momentum distillation." _Thirty-Fifth Conference on Neural Information Processing Systems_. 2021.
> > [^2] Wang, Zirui, et al. "SimVLM: Simple visual language model pre-training with weak supervision." _arXiv preprint arXiv:2108.10904_ (2021).
> > [^3] Dou, Zi-Yi, et al. "An Empirical Study of Training End-to-End Vision-and-Language Transformers." _arXiv preprint arXiv:2111.02387_ (2021).

---

### Official Review · Reviewer_Rnvh · 2021-11-03

**Correctness:** 3
**Technical Novelty And Significance:** 2
**Empirical Novelty And Significance:** 3
**Recommendation:** 6
**Confidence:** 4

**Main Review:**

Strengths:
The paper revisits a broad range of vision and language tasks (VQA, VLN, SNLI-VE, Image Captioning) and evaluates the would-be performance for previous state-of-the-art methods by replacing their visual backbones with CLIP-ResNet/ViT. The diversity of the tasks considered and the two scenarios explored in the paper (straightforward fine-tuning and V&L pre-training) illustrate the strengths of the CLIP-based encoders and highlight some of their limitations.

Weaknesses:
The motivation for all V&L experiments looks weak because it stems from the following single observation “if cast VQA 2.0 into a zero-shot image-to-text retrieval task, we only observe chance performance. Thus, we propose to integrate CLIP’s visual encoder with previous V&L models.” The prompt engineering considered for VQA is not persuasive and does not imply the same straightforward introduction of CLIP-based backbones to VLN / Image Captioning tasks. Also, I think it is necessary to explain the “chance performance” for VQA, at least for the binary yes/no question type. E.g., does 0.037 mean that one can simply flip the model’s answers?

“Unfreezing” the Visual Backbone helps if done correctly – it is a well-known fact supported by BUTD-Res101 “unfreezing” experiment in the paper. However, CLIP-Res50 benefits more from pre-training than BUTD-Res101. This observation requires further elaboration.

Most of the models considered in their respective tasks seem outdated as of 2021 (e.g., Pythia 2019, MCAN 2019). At the same time, the authors avoid direct comparison to the methods that require large-scale “supervised” or “self-supervised” pre-training, e.g., VLN-BERT, although CLIP itself can be considered as such.

The authors may want to include findings on localization ability of transformers from the following papers in the later versions of the manuscript:
-	Do Vision Transformers See Like Convolutional Neural Networks? arxiv 2021
-	Dynamic Head: Unifying Object Detection Heads with Attentions, CVPR 2021
-	Pyramid Vision Transformer: A Versatile Backbone for Dense Prediction without Convolutions, ICCV 2021


**Summary Of The Paper:**

The paper utilizes recently proposed CLIP-ResNet/ViT visual encoders instead of the standard backbones to conduct a large-scale empirical study for several vision and language tasks. The experimental setup aims to answer the question in the title of this paper by conducting 1) the task-specific fine-tuning of existing V&L models with CLIP-based visual backbone and 2) performing full-scale V&L pre-training.
The resulting performance in VQA, Visual Entailment, V&L Navigation suggests that CLIP is a viable alternative to the existing visual representations (e.g., ResNet-based models pre-trained on ImageNet).

**Summary Of The Review:**

It’s not surprising that the CLIP-based vision backbone - after being trained on 400M image-text pairs - can outperform ImageNet-trained baselines in vision and language tasks, especially given its zero-shot performance illustrated in the original work.
The presented results can be helpful for the research community, although the answer to the question posed in the title remains largely open. The outcomes of experiments vary from task to task, with no appropriate discussion followed even for the difference between CLIP-ResNet / CLIP-ViT-B features. As it reads in the Conclusion section: “Analyses from different perspectives explain certain intriguing phenomena and offer new directions for future V&L research.”

---

> ### Author Response · Authors · 2021-11-22
> **Response to Reviewer Rnvh**
>
> We thank the reviewer for their feedback and suggestions.
>
> >1: discussion about CLIP-ResNet versus CLIP-ViT features.
>
> We thank the reviewer for pointing us towards [1, 2, 3] regarding the localization issues of vanilla vision transformers, which we cite in our revised manuscript.
> We add more discussion and results both quantitatively and qualitatively. (see the response to reviewer ZXD1 “why does the visual transformer model of CLIP exhibit worse performance? ”)
>
>
> >2: “explanation towards the ‘chance performance’ for VQA of CLIP zero-shot performance … for the binary yes/no question type”
>
> The reason that the model still get ~0 performance (and not 50% accuracy) for Yes/No questions is because at test time, the model does not know which questions are yes/no questions. When a question is a yes/no question, following standard practice, we still consider all 3129 candidate answers. So the problem is still a 3129-way classification problem and not a binary classification problem and the chance performance is still 0.00031.
>
> >3: “most of the models … seem outdated as of 2021 (e.g., Pythia 2019, MCAN 2019).”
>
> Our results with Pythia and MCAN are intended as a controlled experiment to test VQA performance without V&L pre-training. Since 2019, on VQA, V&L pre-training has gained popularity. Thus, on VQA, we provide the comparison with the most-updated SotA models such as UNITER, Oscar, and VinVL in Table 6, where we allow V&L pre-training. Our method surpasses them all.
>
>
> >4. “the authors avoid direct comparison to the methods that require large-scale “supervised” or “self-supervised” pre-training, e.g., VLN-BERT ....”
>
> Our main claim is that we can power V&L models with CLIP as the visual backbone. Therefore, our experiment focuses on comparing various visual backbones and demonstrates that our design outperforms or is competitive with other **up-to-date visual encoders** in the context of V&L. Comparing with the SOTA V&L models (e.g., VLN-BERT) on specific V&L task may be out of the scope of this paper. In fact, our approach is flexible and capable of integrating into various V&L models. Our approach can indeed be integrated into VLN-BERT to become VLN-BERT + CLIP-ViL. However, due to resource constraints, we defer these experiments to future work. To support this effort, we will release our source code to allow the researcher and practitioners to easily integrate CLIP with large V&L models.
>
> In the current table, we augment a more affordable model EnvDrop to be EnvDrop + CLIP-ViL and we list the performance of VLN-BERT. We find that CLIP-ViL can empower a “outdated” model EnvDrop to become competitive with VLN-BERT, which shows the efficacy of our method.
>
> >5. “CLIP-Res50 benefits more from pre-training than BUTD-Res101. This observation requires further elaboration..”
>
> Thanks for the suggestion. We have revised our presentation on Table 8. Our main finding is that V&L pre-training brings performance improvement for both BUTD and CLIP models. For BUTD, V&L pre-training brings the performance from 64.70 to 72.42; while for CLIP, V&L pre-training brings the performance from 64.66 to 73.92. The two models are V&L pre-trained on the same data.
> Interestingly, we find that CLIP models benefit more from V&L pre-training.
>
> Our conjecture is that the additional benefit could come from unfreezing the visual backbone. Because of technical difficulty in fine-tuning the object detector, most V&L models rely on frozen region-based encoders (BUTD). But for grid-features such as CLIP, we can easily fine-tune the visual backbone and could potentially aid CLIP to adapt to the pre-training task. We are currently running experiments using (frozen CLIP w/ V&L Pretraining) to verify this hypothesis, and will update the results in later versions.
>
> *We hope that our response and revision address your concerns and questions. We are happy to provide discussion if you have any further concerns or comments.*

---

### Decision · Program_Chairs · 2022-01-20

**Decision:**

Accept (Poster)

**Comment:**

Reviewers are in agreement that this work is a useful, clear, documentary piece of work that shows the utility of CLIP on a number of popular V+L tasks.  There is a somewhat persistent concern that simply demonstrating that a stronger visual encoder leads to improvements downstream is not an insightful result on which the community can build.